# COVID-19 in-hospital mortality and mode of death in a dynamic and non-restricted tertiary care model in Germany

Siegbert Rieg[1]*, Maja von Cube[2], Johannes Kalbhenn[3], Stefan Utzolino[4], Katharina Pernice[5], Lena Bechet[1], Johanna Baur[6,7], Corinna N. Lang[6,7], Dirk Wagner[1], Martin Wolkewitz[2], Winfried V. Kern[1], Paul Biever[6,7], on behalf of the COVID UKF Study Group[¶]

1 Division of Infectious Diseases, Department of Medicine II, Medical Center–University of Freiburg, Faculty of Medicine, University of Freiburg, Freiburg, Germany, 2 Institute of Medical Biometry and Statistics, Faculty of Medicine and Medical Center, University of Freiburg, Freiburg, Germany, 3 Department of Anesthesiology and Intensive Care Medicine, Medical Center–University of Freiburg, Faculty of Medicine, University of Freiburg, Freiburg, Germany, 4 Department of General and Visceral Surgery, Medical Center–University of Freiburg, Faculty of Medicine, University of Freiburg, Freiburg, Germany, 5 Department of Cardiovascular Surgery, Heart Center, Medical Center–University of Freiburg, Faculty of Medicine, University of Freiburg, Freiburg, Germany, 6 Department of Medicine III (Interdisciplinary Medical Intensive Care), Medical Center–University of Freiburg, Faculty of Medicine, University of Freiburg, Freiburg, Germany, 7 Department of Cardiology and Angiology I, Heart Center Freiburg University, Medical Center–University of Freiburg, Faculty of Medicine, University of Freiburg, Freiburg, Germany

¶ Members and details of the COVID UKF Study Group are provided under Acknowledgments
* siegbert.rieg@uniklinik-freiburg.de

## Abstract

### Background

Reported mortality of hospitalised Coronavirus Disease-2019 (COVID-19) patients varies substantially, particularly in critically ill patients. So far COVID-19 in-hospital mortality and modes of death under state of the art care have not been systematically studied.

### Methods

This retrospective observational monocenter cohort study was performed after implementation of a non-restricted, dynamic tertiary care model at the University Medical Center Freiburg, an experienced acute respiratory distress syndrome (ARDS) and extracorporeal membrane-oxygenation (ECMO) referral center. All hospitalised patients with PCR-confirmed SARS-CoV-2 infection were included. The primary endpoint was in-hospital mortality, secondary endpoints included major complications and modes of death. A multistate analysis and a Cox regression analysis for competing risk models were performed. Modes of death were determined by two independent reviewers.

### Results

Between February 25, and May 8, 213 patients were included in the analysis. The median age was 65 years, 129 patients (61%) were male. 70 patients (33%) were admitted to the intensive care unit (ICU), of which 57 patients (81%) received mechanical ventilation and 23

**Data Availability Statement:** Due to the German Federal Data Protection Act (Bundesdatenschutzgesetz) and the fact that the Institutional Review Board of the University

Medical Center Freiburg granted publication of only anonymyzed data, inclusion of the complete dataset is not possible. The R-code for data analysis in included in the Supporting Information. Requests for the anonymized dataset from interested researchers can be sent to the Division of Infectious Diseases, Department of Medicine II, Medical Center – University of Freiburg, Germany (info@if-freiburg.de).

**Funding:** MVC was funded by the EQUIP programme of the Faculty of Medicine, University of Freiburg (https://www.med.uni-freiburg.de/de/forschung/karrierewege/equip/equip). The funders had no role in study design, data collection and analysis, decision to publish, or preparation of the manuscript.

**Competing interests:** The authors have declared that no competing interests exist.

patients (33%) ECMO support. Using multistate methodology, the estimated probability to die within 90 days after COVID-19 onset was 24% in the whole cohort. If the levels of care at time of study entry were accounted for, the probabilities to die were 16% if the patient was initially on a regular ward, 47% if in the intensive care unit (ICU) and 57% if mechanical ventilation was required at study entry. Age ≥65 years and male sex were predictors for in-hospital death. Predominant complications–as judged by two independent reviewers–determining modes of death were multi-organ failure, septic shock and thromboembolic and hemorrhagic complications.

## Conclusion

In a dynamic care model COVID-19-related in-hospital mortality remained very high. In the absence of potent antiviral agents, strategies to alleviate or prevent the identified complications should be investigated. In this context, multistate analyses enable comparison of models-of-care and treatment strategies and allow estimation and allocation of health care resources.

## Introduction

The current SARS-CoV-2 pandemic is a public health emergency of international concern, which poses immense challenges on health care systems [1]. Although modulated by host factors like age and comorbidities, overall about 10–15% of SARS-Cov-2 infected patients require hospitalisation and 20–30% of hospitalised patients develop critical or life-threatening COVID-19 manifestations [2]. Reported mortality rates of COVID-19 patients are in the range of 20–40% [1,3–5] for hospitalised patients and 30–88% for critically-ill or ICU patients with substantial differences between countries and regions [3–10]. Several reasons may account for the observed wide range of these estimates. Referral strategies to the hospital may differ. A high local COVID-19 incidence may put pressure on health care systems leading to restrictions in care with the need to triage patients, and possibly results in high numbers of infected health care workers. Moreover, intensive care unit (ICU) and therefore ventilation and extracorporeal membrane-oxygenation (ECMO) capacities may substantially vary, which may influence admission strategies and decisions on treatment withdrawal.

Compared to neighbouring countries, in Germany the SARS-CoV-2 pandemic started later, providing the health care system and particularly the inpatient sector with valuable time to prepare for a rising case load. The Freiburg University Medical Center, a center with profound expertise in ARDS treatment and ECMO support, formed a Coronavirus task force at the end of January 2020. In the following weeks a COVID-19 dynamic care model was developed and implemented. These preparations together with a relatively high SARS-CoV-2 testing capacity and early lock-down strategies in Germany yielded a situation, in which regional treatment capacities were sufficient at any stage of the pandemic and at any level of care.

We hypothesised that this constitutes a unique opportunity to study the COVID-19-related morbidity and mortality in patients requiring hospitalisation in a setting of non-restricted care. Here we briefly outline the implemented dynamic care model and summarise the corresponding outcomes. Specific aims of the study are i.) to assess COVID-19-related in-hospital mortality in a dynamic and non-restricted care model at an ARDS and ECMO referral center; ii.) to define major complications and modes of death in a setting of extended care with

maximum supportive therapy; and iii.) to propagate and stimulate reporting of clinical studies in COVID-19 research using multistate models.

## Methods

### Study design, setting and participants

The current study constitutes a *post hoc* analysis of data collected within a retrospective cohort study conducted at the University Medical Center Freiburg. This 1,600-bed tertiary care institution serves the southwest region of the German state of Baden-Württemberg and is one of the largest ARDS and ECMO referral centers in Germany. All hospitalised patients with detection of SARS-CoV-2 using PCR in a respiratory sample between February 25 and May 8, 2020 were eligible and included. The last day of follow-up that was included was June 19.

Beginning in January 2020 the Coronavirus task force at the University Medical Center Freiburg developed a dynamic care model for COVID-19 patients (outlined in **S1 Fig**). Patients were treated on COVID-19 regular wards, COVID-19 intermediate care and intensive care units (ICU) run by different departments. Patients were followed during their hospital stay by Infectious Diseases (ID) physicians performing daily COVID-19 rounds. The measures implemented in the COVID-19 response, the evolution of the peak incidences in the region and the corresponding number of admissions in our center are shown in S2 Fig.

### Variables collected and definitions

Demographic variables, comorbidities, diagnostic procedures and data on treatment modalities, complications and outcome were extracted by reviewing the admission, transfer and discharge reports and the electronic patient record. Patients were followed until hospital discharge or death.

Comorbidities were recorded in the following eight categories: lung disease (COPD or other chronic pulmonary disease), heart disease (coronary artery disease/ischemic cardiomyopathy or heart failure NYHA II-IV), diabetes mellitus, chronic liver disease (Child B or C), active malignancy, primary or secondary immunodeficiency (the latter being immunosuppressive drugs incl. corticosteroids of ≥20mg/day prednisolone-equivalent), obesity (body mass index [BMI]>30kg/m$^2$) and neurological disease (dementia, stroke or Parkinson's disease). For Cox regression analysis patients were divided into the groups 'no comorbidity' and 'at least one comorbidity' present. Hospital-acquired COVID-19 was assumed in the setting of prolonged hospitalisation and if contact tracing yielded contact with other COVID-19 patients or health-care workers in the hospital as the only relevant exposure.

A thorough case review by two independent investigators (intensivists [ICU patients] or ID physicians) concerning complications and modes of death was performed for all patients. All discrepancies between the two reviewers were reviewed and resulted in an additional assessment by a third investigator in order to obtain a final decision.

Classification of ARDS severity was performed according to the Berlin Definition [11]. Indication for ECMO support was in accordance with the guidelines of the Extracorporeal Life Support Organization (ELSO) [12] and did not deviate from usual indications. Multi-organ failure (MOF) was defined as combination of two or more severe organ system dysfunctions. Predominant terminal organ failure during dying process was defined as severe organ dysfunction that either resulted directly in patient´s death or in withdrawal of life support. Concerning the categories 'Life support in dying process' and 'Involvement of COVID-19', patients were allocated to one category. Reviewers designated each death as either ‚related to COVID-19' or ‚unrelated to COVID-19'.

### Ethical consideration

The study and data collection were approved by the Institutional Review Board of the University Medical Center Freiburg (348/20) and was registered in the German Clinical Trials Register (identifier DRKS00021775). We followed the ethical standards set by the Helsinki Declaration of 1964, as revised in 2013, and the research guidelines of the University of Freiburg. The Institutional Review Board of the University Medical Center Freiburg considered the collection of routine data as evaluation of service and waived the need for written informed consent. The Institutional Review Board approved the publication of anonymized data.

### Statistical analysis

The primary endpoint was in-hospital mortality. Secondary endpoints included major complications and modes of death. Baseline epidemiological and clinical characteristics, complications and outcomes of patients with and without ICU stay were compared using the t-test or Mann-Whitney-U-test for continuous variables and the $\chi^2$ test or Fisher's exact test for categorical variables.

We performed a Markovian multistate analysis [13] to investigate the mean length of hospitalisation, the mean duration of mechanical ventilation (MV) and ECMO as well as the risks of death and discharge. Multistate model analysis has not only the major advantage that the time dyamics of a patient's disease progression are taken into account but also that multiple events are studied simultaneously. The model is shown in S3 Fig. The statistical methodology and required assumptions are outlined in detail in [14]. The multistate model accounts for the states hospitalisation in a 'regular ward', 'ICU', 'MV', 'ECMO' as well as 'discharge alive' and 'death'. Patients entered the study at the time of hospitalisation due to COVID-19 or at the time of a positive SARS-CoV-2-PCR (in hospital-acquired COVID-19 cases) and were under observation until discharge or death.

For the risk factor analysis, we used a competing risks model to study effects on the time from hospitalisation to death in the hospital. To avoid collider bias, in this model the different states of hospitalisation (regular ward, ICU, MV, ECMO) were not differentiated. First, we estimated cause-specific hazard ratios for death and discharge. These gave information on both direct and indirect effects on the risk of in-hospital death. Then, we estimated the subdistribution hazard ratio of death using a Fine and Gray model. The subdistribution hazard ratio quantifies the effect of risk factors on the absolute risks (rather than the rates) thereby combining the direct and indirect effects found in the cause-specific analysis. Statistical significance was determined at $p < 0.05$. All analyses were performed with R Version 4.0.2.

## Results

### Epidemiological and clinical characteristics

A total of 213 COVID patients were included in the study (Table 1). The median age was 65 years, 129 patients (61%) were male. Fifty cases (23%) were considered to be hospital-acquired infections. While 56 patients (26%) were without significant comorbidities, 79 patients (37%) reported one, and 78 patients (37%) two or more comorbidities, with coronary artery disease/ischemic cardiomyopathy (21%), diabetes mellitus (20%) and obesity (BMI>30mg/m$^2$, 24%) being the most prevalent diseases. The median time from onset of symptoms to hospitalisation was 6 days. Overall 27 patients (13%) were ICU-referrals from regional hospitals due to complex respiratory or ARDS management and/or the need of ECMO support. During hospitalisation 70 patients (33%) were admitted to the ICU (median SAPS2-score of 46, median Horovitz-index on day 1 of ICU admission 110), of which 57 patients (81%) received invasive

**Table 1. Epidemiological and clinical characteristics of 213 COVID-19 patients with and without ICU care.**

| Parameter | All patients n = 213 | Patients with Non-ICU care n = 143 | Patients with ICU care n = 70 | p-value |
|---|---|---|---|---|
| Age | 65 (54–79;25) | 65 (53–80;27) | 65 (59–76;17) | 0.86 ** |
| Sex male | 129 (61) | 77 (54) | 52 (74) | 0.004 * |
| Time from clinical onset of symptoms to admission (n = 137) | 6 (3–9;6) | 5 (2–9;7) | 7 (4–11;7) | 0.04** |
| NEWS2-Score (n = 172) | 7 (3–10; 7) | 5 (3–8; 5) | 10 (8–12; 4) | <0.0001** |
| **Comorbidities** | | | | |
| COPD | 13 (6) | 6 (4) | 7 (10) | 0.10* |
| Coronary artery disease/ischemic cardiomyopathy | 45 (21) | 29 (20) | 16 (23) | 0.67* |
| Malignancy/neoplasm | 29 (14) | 20 (14) | 9 (13) | 0.82* |
| Chemotherapy within last 3 months | 9 (4) | 6 (4) | 3 (4) | 0.98* |
| Primary or secondary immunodeficiency incl. immunosuppressive medication | 26 (12) | 20 (14) | 6 (9) | 0.26* |
| Diabetes mellitus | 43/158 (20) | 29/92 (20) | 14/66 (20) | 0.96* |
| Obesity (BMI >30 kg/m$^2$) | 38 (24) | 20 (22) | 18 (27) | 0.42* |
| Number of comorbid conditions | | | | |
| No comorbid condition | 56 (26) | 38 (27) | 18 (26) | 0.95* |
| 1 comorbid condition | 79 (37) | 52 (36) | 27 (39) | |
| ≥2 comorbid conditions | 78 (37) | 53 (37) | 25 (36) | |
| **Laboratory investigations on admission** | | | | |
| Lymphocytes [per μl] (n = 125) Norm: 800–3.000 per μl | 830 (510–1170; 660) | 870 (560–1170; 610) | 710 (470–1110; 640) | 0.21** |
| Thrombocytes [×10$^3$/μl] (n = 207) Norm: 176–391 ×10$^3$/μl | 190 (150–253; 103) | 186 (150–235; 85) | 217 (150–286; 136) | 0.11** |
| CRP [mg/l] (n = 204) Norm: <5 mg/l | 68 (22–134; 112) | 36 (12–96; 84) | 137 (81–226; 145) | <0.0001** |
| PCT [ng/ml] (n = 182) Norm: <0,05 ng/ml | 0,15 (0,08–0,45; 0,37) | 0,11 (0,06–0,19; 0,13) | 0,47 (0,21–1,47; 1,26) | <0.0001** |
| IL-6 [pg/ml] (n = 147) Norm: <7 pg/ml | 50 (22–146; 124) | 32 (16–51; 35) | 175 (77–729; 652) | <0.0001** |
| D-dimers [mg/l FEU] (n = 97) Norm: <0,5 mg/l | 1,4 (0,6–4,6; 4) | 1,0 (0,51–1,8; 1,3) | 2,3 (1,4–11,9; 10,5) | <0.0001** |
| Troponin T [ng/l] (n = 127) Norm: <14 ng/l | 16 (7–39; 32) | 10 (6–30; 24) | 29 (12–61; 49) | 0.003** |
| **Medical treatment** | | | | |
| Intravenous antibiotics | 131 (62) | 66 (46) | 65 (93) | <0.0001* |
| Lopinavir/ritonavir | 54 (25) | 17 (12) | 37 (53) | <0.0001* |
| Hydroxychloroquine/chloroquine | 92 (43) | 39 (27) | 53 (76) | <0.0001* |
| Tocilizumab | 7 (3) | 1 (1) | 6 (9) | 0.006*** |
| **Outcomes (at end of follow-up)** | | | | |
| Discharged, n (%) | 161 (69) | 124 (87) | 37 (53) | <0.0001 * |
| Death in hospital, n (%) | 51 (23) | 18 (13) | 33 (47) | |
| Still hospitalised, n (%) | 1 (0,5) | 1 (1) | 0 (0) | |

Data are median and interquartile range (IQR) or numbers (%).

*$\chi^2$-test

**Mann-Whitney U test

***Fisher's exact test.

ICU, intensive care unit; NEWS2, National Early Warning Score 2; BMI, body mass index; COPD, chronic obstructive pulmonary disease; CRP, C-reactive protein; PCT, procalcitonin; IL-6, Interleukin-6; Norm, normal range.

MV (median duration 17 days), and 23 patients (33%) needed ECMO support (median duration 11 days, range 1–68 days) (Table 2). Medical treatment included lopinavir/ritonavir (54 patients), hydroxychloroquine (92 patients), and remdesivir (1 patient). Seven patients received tocilizumab. 161 out of 213 patients were discharged alive and 51 patients died. Of

**Table 2. Management and complications of 70 ICU patients with COVID-19.**

| Characteristics of ICU patients | All patients n = 70 | Survivors n = 37 | Non-Survivors n = 33 | p-value |
|---|---|---|---|---|
| Age | 64.5 (59–76) | 61 (54–70) | 70 (61–78) | 0.01** |
| Direct ICU referrals | 27 (39) | 16 (43) | 11 (33) | 0,40* |
| Blood type 0 | 15/65 (23) | 9/33 (27) | 6/32 (19) | 0.41* |
| Blood type A | 38/65 (59) | 20/33 (61) | 18/32 (56) | 0.72* |
| **Disease severity upon ICU admission** | | | | |
| SAPS2-score (d1) | 46 (40–52) | 45 (31–50) | 49 (45–55) | 0,005** |
| No ARDS or mild ARDS | 6 (9) | 4 (11) | 2 (6) | 0,09* |
| Moderate ARDS | 27 (39) | 18 (49) | 9 (27) | |
| Severe ARDS | 37 (53) | 15 (41) | 22 (67) | |
| Horovitz-Index (lowest in first 24h after ICU admission) | 110 (82–126) | 114 (88–137) | 96 (79–116) | 0,13** |
| **ICU Management** | | | | |
| High-flow nasal cannula | 30 (43) | 20 (54) | 10 (30) | 0,05* |
| Non-invasive mechanical ventilation | 30 (43) | 15 (41) | 15 (46) | 0,68* |
| High-flow nasal cannula or non-invasive mechanical ventilation (and no invasive mechanical ventilation) | 6 (9) | 5 (14) | 1 (3) | 0,20*** |
| Invasive mechanical ventilation | 57 (81) | 28 (76) | 29 (88) | 0,23*** |
| Median length of invasive mechanical ventilation, days | 17 (8–32) | 19.5 (9–40) | 15 (7–22) | 0,13** |
| Tracheostomy | 26 (37) | 17 (46) | 9 (27) | 0,11* |
| ECMO | 23 (33) | 9 (24) | 14 (42) | 0,11* |
| Length of ECMO treatment, days | 11 (7–21) | 9 (8–23) | 12 (4–22) | 0,79** |
| ECMO cannulation in external hospital | 9/23 (39) | 3/9 (33) | 6/14 (43) | >0,999*** |
| ECMO weaning successful | 12/23 (52) | 9/9 (100) | 3/14 (21) | 0,0003*** |
| Veno-arterial ECMO or left ventricular unloading (Impella®) | 4/23 (17) | 0 | 4/14 (29) | 0,13*** |
| Prone-positioning | 43 (61) | 21 (57) | 22 (67) | 0,40* |
| Number of prone-positionings per patient | 9 (5–13) | 8.5 (5–13) | 9.0 (6–14) | 0,85** |
| Prone-positioning during ECMO | 19/23 (83) | 8/9 (89) | 11/14 (79) | >0,999*** |
| Repeated neuromuscular blockade | 11 (16) | 4 (11) | 7 (21) | 0,33*** |
| Inhaled nitric oxide | 6 (9) | 4 (11) | 2 (6) | 0,68*** |
| **Complications** | | | | |
| Pulmonary embolism (CT-verified) | 16 (23) | 10 (27) | 6 (18) | 0,38* |
| Central pulmonary embolism | 5/16 (31) | 3/10 (30) | 2/6 (33) | >0,999*** |
| Segmental/subsegmental pulmonary embolism | 16/16 (100) | 10/10 (100) | 6/6 (100) | >0,999*** |
| Acute kidney injury with need of renal replacement therapy | 26 (37) | 12 (32) | 14 (42) | 0,39* |
| Replacement of renal replacement system due to thrombosis (at least once) | 11/26 (42) | 6/12 (42) | 5/14 (50) | 0,46* |
| ECMO system or ECMO pump replacement system due to thrombosis (at least once) | 12/23 (52) | 5/9 (56) | 7/14 (50) | >0,999*** |
| Intracerebral bleeding (CT-verified) | 11 (16) | 5 (11) | 6 (16) | 0,59* |
| Intracerebral bleeding w/o ECMO | 6/47 (13) | 3/28 (11) | 3/19 (18) | 0,67*** |
| Ischemic stroke | 9 (13) | 3 (8) | 6 (11) | 0,29*** |
| Ischemic stroke w/o ECMO | 4/47 (9) | 2/28 (7) | 2/19 (11) | >0,999*** |
| Cardiac arrest with ROSC | 6 (9) | 1 (3) | 5 (15) | 0,09*** |
| Pulmonary bleeding | 8 (11) | 3 (8) | 5 (15) | 0,46*** |
| Pneumothorax | 12 (17) | 5 (14) | 7 (21) | 0,39* |
| Septic shock | 43 (61) | 17 (46) | 26 (79) | 0,005* |
| Cardiogenic shock | 13 (19) | 5 (14) | 8 (24) | 0,25* |
| Hemorrhagic shock | 9 (13) | 4 (11) | 5 (15) | 0,73*** |
| Pulmonary bacterial superinfection† | 26 (37) | 15 (41) | 11 (33) | 0,53* |
| Positive blood cultures | 28 (40) | 18 (49) | 10 (30) | 0,12* |

(*Continued*)

**Table 2.** (Continued)

| Characteristics of ICU patients | All patients n = 70 | Survivors n = 37 | Non-Survivors n = 33 | p-value |
|---|---|---|---|---|
| Positive blood cultures (without typical contaminants of skin flora) | 16 (23) | 9 (24) | 7 (21) | 0,76* |
| Aspergillus positive respiratory samples with initiation of antifungal therapy | 6 (9) | 1 (3) | 5 (15) | 0,09*** |

Data are median and interquartile range (IQR) or numbers (%).

*$\chi^2$-test

**Mann-Whitney U test

***Fisher's exact test.

ICU, intensive care unit; ARDS, acute respiratory distress syndrome; CT, computed tomography scan; ECMO, extracorporeal membrane-oxygenation.

† Positive respiratory samples with *Staphylococcus aureus*, *Streptococcus pneumoniae* or Gram-negative bacteria (*Escherichia coli*, *Klebsiella pneumoniae*, *Pseudomonas aeruginosa*, *Proteus mirabilis*, *Enterobacter cloacae*, *Citrobacter freundii*, *Serratia marcescens*) with initiation of antibacterial treatment.

the latter, 32 deaths occurred in the ICU (one death after ICU discharge) and 18 deaths on regular wards. At the end of follow-up, one patient, though recovered from COVID-19, was still hospitalised on a regular ward for treatment of an underlying malignancy.

## Multistate model analysis

Considering all 213 patients in the described dynamic tertiary care model, the population averaged probability to have died 90 days after hospitalisation with COVID-19 was 23.9%. The chance for being discharged alive was 75,6%. There was a 0.5% chance to still be in the hospital after 90 days. A stacked probability plot illustrating the probabilities of COVID-19 patients to be in specific states (regular ward, ICU, MV, ECMO, discharged alive or dead) over the course of time is depicted in Fig 1. Moreover, the plot illustrates the population averaged mean duration spent in each state/level of care. These correspond to the coloured area between two curves.

By accounting for the levels of care when entering the study, i.e. regular ward, ICU, MV, the multistate model allows for an estimation of the approximate length of hospital stay and the probability to be discharged alive or to die at different levels of care. A patient that was first admitted to a regular ward stayed on average 13.6 days in the hospital, 0.8 days in the ICU, 1.4 days with MV and 0.2 days with MV and ECMO within a total stay of 90 days (Fig 2 and S4

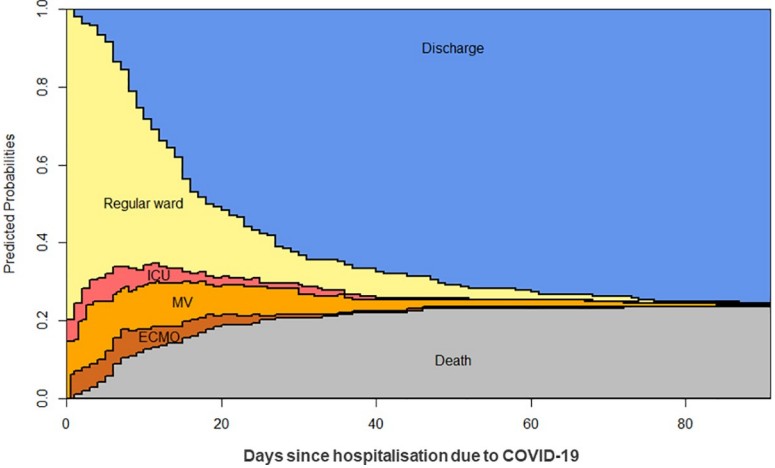

**Fig 1.**

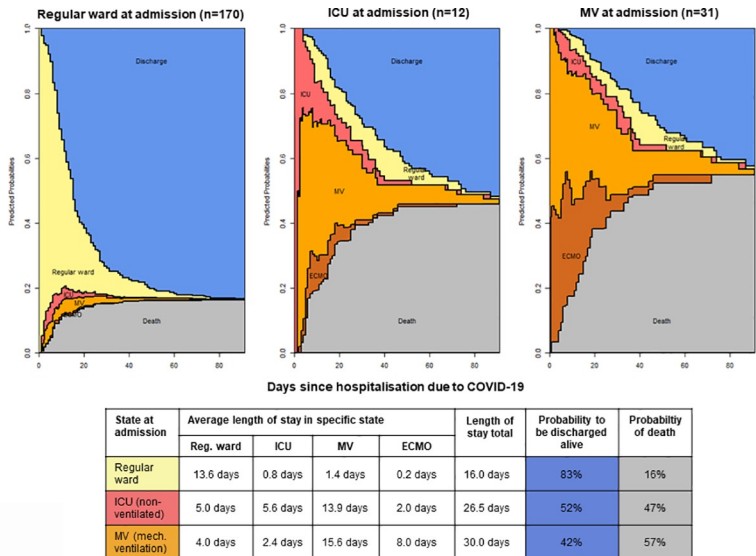

**Fig 2.**

Fig). The probability to be discharged alive for patients starting in the ‚regular ward'-state was 83%, the probability to die was 16%. In contrast, a patient that was admitted to the ICU needed 21.5 days in the ICU, 13.9 days of these with MV and 2.0 days with ECMO. The probability to be discharged alive in the following 90 days was 52%, the probability to die was 47%. Patients that directly required MV stayed 23.6 days on MV, and 8.0 days of these with ECMO. Once MV was no longer required, the patient stayed on average 2.4 more days in the ICU and another 4.0 days on the regular ward. The chances to be discharged alive were only 42%.

## Multivariable cause-specific Cox regression analysis

The multivariable regression analysis constitutes a competing risks model with the endpoint in-hospital death and the competing risk discharge alive. According to the cause-specific Cox regression older patients have a higher death hazard (HR 3.45, 95% CI 1.49–7.98, for patients 65–74 years of age, and HR 3.56, 95% CI 1.74–7.30 for ≥75 years-aged patients). Additionally, we found that the discharge hazard is significantly decreased for males (HR 0.68, 95% CI 0.50–0.94) (Table 3). A higher number of comorbid conditions was not significantly associated with altered death or discharge hazards.

In the Fine and Gray model yielding subdistribution hazard ratios, the probability to die was significantly increased for males (HR 1.90, 95% CI 1.04–3.48) and patients aged 65 years or older (HR 4.16, 95% CI 1.82–9.49 for age group 65–74 years, and HR 4.13, 95% CI 2.05–8.32 for ≥75 years of age). For males the decreased discharge hazard leads to a prolonged length of stay and therefore increased the risk of death in the hospital. The increased death risk for patients older than 65 is explained by a direct effect on the death hazard. Stacked probability plots (S5–S8 Figs and S1 Data) stratified respectively by age, sex, the presence of comorbidities, immunodeficiency and malignancy/neoplasm illustrate in detail the effect of these risk factors not only on mortality, but also on the six states of the multistate model.

## Complications and presumed modes of death

According to the individual case review, ICU patients (both, survivors and non-survivors) suffered from a multitude of complications (Table 2), the four dominant ones being septic shock

**Table 3. Multivariable Cox regression analysis.**

| Model/Analysis | Multivariable Cox regression | | | | | |
|---|---|---|---|---|---|---|
| Endpoint | Discharge | | | Death | | |
| Variable | Hazard ratio | 95% CI | p-value | Hazard ratio | 95% CI | p-value |
| Sex male[1] | 0.68 | 0.50–0.94 | **0.020** | 1.37 | 0.74–2.54 | 0.310 |
| Age 65–74 years[2] | 0.63 | 0.37–1.06 | 0.079 | 3.45 | 1.49–7.98 | **0.004** |
| Age ≥ 75 years[2] | 0.71 | 0.49–1.02 | 0.067 | 3.56 | 1.74–7.30 | **0.001** |
| Hospital-acquired COVID-19[3] | 0.73 | 0.48–1.12 | 0.155 | 0.91 | 0.45–1.84 | 0.790 |
| Comorbidities present (≥1)[4] | 0.87 | 0.60–1.25 | 0.442 | 1.30 | 0.61–2.79 | 0.494 |
| Length of stay[5] | 1.00 | 0.99–1.01 | 0.739 | 0.98 | 0.94–1.02 | 0.372 |
| Model/Analysis | Fine and Gray model | | | | | |
| Endpoint | | | | Death | | |
| Variable | | | | Subdistribution hazard ratio | 95% CI | p-value |
| Sex male[1] | | | | 1.90 | 1.04–3.48 | 0.03 |
| Age 65–74 years[2] | | | | 4.16 | 1.82–9.49 | <0.001 |
| Age ≥ 75 years[2] | | | | 4.13 | 2.05–8.32 | <0.001 |
| Hospital-acquired COVID-19[3] | | | | 1.18 | 0.60–2.34 | 0.59 |
| Comorbidities present (≥1)[4] | | | | 1.25 | 0.59–2.68 | 0.55 |
| Length of stay[5] | | | | 0.98 | 0.94–1.03 | 0.23 |

[1] Reference: female

[2] reference: age 0–64 years

[3] reference: community-acquired COVID-19

[4] reference: no comorbid condition

[5] reference: 0 days (Previous length of stay was the time from hospital admission to COVID-19 onset, for patients with community acquired COVID-19, the length of stay was 0 days).

in 43 patients (61%), acute kidney injury with the need for renal replacement therapy in 26 of 70 patients (37%), as well as thromboembolic and hemorrhagic complications. Pulmonary embolism was diagnosed in 16 patients (23%). Replacement of extracorporeal devices due to thrombosis had to be performed in 11 of 26 patients (42%) on renal replacement therapy and 12 of 23 patients (52%) on ECMO. Ischemic stroke occurred in 9 of 70 patients (13%). Major hemorrhagic manifestations were intracerebral bleeding in 11 patients (16%) and pulmonary hemorrhage in 8 patients (11%).

As of June 19, 2020, 18 patients died on regular wards. The median age of these patients was 80 years–in accordance to the patients' will, ICU transfer/treatment and MV was withheld in these patients. Death was due to respiratory failure in 12 patients and multi-organ failure in 6 patients.

All but four patients that received ICU care succumbed due to multi-organ failure (Tables 2, 4 and 5). A median of three organ systems were involved with lung failure (32 patients), kidney/renal failure (24 patients), brain injury (17 patients), heart failure (14 patients) and gastrointestinal injury (13 patients, in particular acute mesenteric ischemia) being the predominant terminal organ failures involved. In 21 of 33 patients (63%) septic shock was a critical complication considered to be relevant for multi-organ failure and death. Of 51 patients that died, death was presumed to be secondary to COVID-19 in 30 patients with frailty/comorbidities. Sixteen patients (31%) without relevant comorbidities, i.e. without underlying diseases impacting on life expectancy, died due to COVID-19 or COVID-19-related complications.

**Table 4. Critical terminal organ failure and modes of death in 51 patients with COVID-19.**

| Parameter | Patients who died n = 51 | Patients who died Non-ICU care n = 18 | Patients who died ICU care n = 33 | p-value* |
|---|---|---|---|---|
| **Predominant terminal organ failure during dying process** | | | | |
| Septic shock | 21 (41) | 0 | 21 (63) | 0.001 |
| Multiorgan failure (n> = 2) | 35 (69) | 6 (33) | 29 (88) | 0.001 |
| Failure of 2 organs | 9 (18) | 4 (22) | 5 (15) | 0.03** |
| Failure of 3–4 organs | 16 (31) | 2 (11) | 14 (42) | |
| Failure of >4 organs | 10 (20) | 0 | 10 (30) | |
| Lung failure | 49 (96) | 17 (94) | 32 (97) | >0,999 |
| IMV and ECMO used | 14 (28) | 0 | 14 (42) | <0.0001 |
| IMV used, no ECMO used | 16 (31) | 1 (6) | 15 (46) | |
| No IMV, no ECMO used | 19 (37) | 16 (89) | 3 (9) | |
| Heart failure | 15 (29) | 1 (6) | 14 (42) | 0.009 |
| Kidney injury | 27 (53) | 3 (17) | 24 (73) | 0.0003 |
| Gastro-intestinal injury | 13 (26) | 0 | 13 (39) | 0.002 |
| Liver failure | 9 (18) | 1 (6) | 8 (24) | 0.13 |
| Brain injury any | 20 (39) | 3 (17) | 17 (52) | 0.02 |
| Intracerebral hemorrhage | 5 (10) | 0 | 5 (16) | 0.15 |
| Thrombembolic event and non-cerebral hemorrhage | 11 (22) | 0 | 11 (33) | 0.005 |
| Cardiogenic shock | 7 (14) | 0 | 7 (21) | 0.04 |
| Cardiac arrest—CPR w/o ROSC | 5 (10) | 1 (6) | 4 (12) | 0.64 |
| **Life support in dying process** | | | | |
| Withholding of ICU | 17 (33) | 17 (94) | 0 | <0.0001** |
| Initial ICU therapy, withdrawal in worsening condition | 18 (35) | 0 | 18 (55) | |
| Full care | 16 (31) | 1 (6) | 15 (46) | |
| **Involvement of COVID-19 as jugdeg by two independent reviwers** | | | | |
| Death presumed due to COVID-19 in patients with normal life expectancy | 16 (31)† | 1 (6) | 15 (46) | 0.01** |
| Death presumed due to COVID-19 in patient with frailty/comorbidities | 30 (59) | 15 (83) | 15 (46) | |
| Death presumed due other condition incl. frailty/comorbidities | 5 (10) | 2 (11) | 3 (9) | |

Data are numbers (%).

*Fisher's exact test, except

**$\chi^2$-test.

ECMO, extracorporeal membrane-oxygenation; IMV, invasive mechanical ventilation; CPR w/o ROSC, cardiopulmonary resuscitation without return of spontaneous circulation.

† Mean years of potential life lost (YPLL) per patient (according to current average life expectancy) 13,1 years.

## Discussion

The principal findings of this study are as follows. i.) In the implemented care model yielding non-restricted conditions at an experienced ARDS and ECMO referral center, COVID-19-related in-hospital-mortality remained high at around 25%. ii.) Older age and male sex were independent risk factors for death. iii) In patients requiring ICU care, 1 out of 2 patients died with critical events being lung and multi-organ failure, septic shock, and thromboembolic and hemorrhagic complications. iv.) In the setting of a referral center the average length of stay in the hospital for COVID-19 patients was 16 days if admittance was to a regular ward,

**Table 5. Terminal organ failure and modes of death in 51 patients with COVID-19 as judged by two independent reviewers.**

| Predominant terminal organ failure during dying process | Lung failure and ECMO support | Lung failure and invasive MV (w/o ECMO) | Lung failure (w/o ECMO or MV) | heart failure | Kidney injury | Gastro intestinal failure |
|---|---|---|---|---|---|---|
| Discordance | 0 | 3 | 0 | 10 | 4 | 5 |
| Concordance | 0 | 48 | 51 | 41 | 47 | 46 |
| % concordance after second review | 100 | 94 | 100 | 80 | 92 | 90 |
| Predominant terminal organ failure during dying process | Liver failure | Brain injury | Thrombembolic event and non-cerebral hemorrhage | Septic shock | Cardiogenic shock | CPR w/o ROSC |
| Discordance | 6 | 5 | 3 | 6 | 4 | 3 |
| Concordance | 45 | 46 | 48 | 45 | 47 | 48 |
| % concordance after second review | 88 | 90 | 94 | 88 | 92 | 94 |
| | Life support in dying process | | | Involvement of COVID-19 as jugded by two independent reviewers | | |
| | Withholding of ICU | Withdrawal of ICU therapy | Full care | Death presumed to COVID-19 in patients with normal life expectancy | Death presumed due to COVID-19 in patient with frailty/comorbidities | Death presumed due other condition incl. frailty/ comorbidities |
| Discordance | 0 | 17 | 11 | 7 | 14 | 8 |
| Concordance | 51 | 34 | 40 | 44 | 37 | 43 |
| % concordance after second review | 100 | 67 | 78 | 86 | 73 | 84 |

26.5 days for patients admitted to the ICU, and 30 days in the case of initial MV in hospital. In the latter group 11 days of ECMO support were required.

In the ongoing SARS-CoV-2 pandemic solid estimates on patient outcomes such as mortality and major complications are pivotal and strongly required by medical and social institutions, yet difficult to generate [15]. Although COVID-19 studies are published at unprecedented frequency and speed, comparability of studies is hampered by the use of different study designs, varying standards of reporting and the statistical approaches used. So far, the majority of studies, particularly those in critically-ill or ICU patients, reported on preliminary in-hospital mortality rates, as 23–72% of patients were still hospitalised at the time of reporting [5,7–10].

We believe our study provides superior estimates on mortality, complications and length of stay, as different study set up and analytical approaches compared to previous studies were employed. First, by implementing a dynamic care model, we excluded that the need to triage patients, or the availability of limited ICU capacities impacted on mortality rate in a major way. Moreover, given the experience of a large interdisciplinary ARDS and ECMO referral center together with a highly active ID service, the conditions to manage critically ill COVID-19 patients with severe pneumonia and development of ARDS adhered to highest international standards. However, the COVID-19 related in-hospital mortality rate of 24% overall, of 47% in the ICU subgroup and of 57% in the MV subgroup remained substantial even under maximal respiratory support with prolonged provision of ECMO and other advanced therapies including prone-positioning. Of note, about one third of patients that died were without relevant comorbidities and were believed to have a normal life expectancy prior to SARS-CoV-2 infection.

The identified risk factors for death, namely age and male sex, are in line with findings of published studies. Interestingly, application of a competing risk model identified male sex to be associated with a decreased discharge hazard, thereby contributing indirectly to an increased risk of death. Comorbidities were either equally distributed or more often prevalent in the ICU subgroup, with the only exception of immunodeficiency, which was more frequent in the Non-ICU group. Although not adjusted to other factors, our results point towards a comparable COVID-19-related mortality in patients with and without immunodeficiency.

The present study comprises 70 ICU patients, including 23 patients with ECMO support. It is the first study with a completed follow-up, as all patients were discharged from the ICU. The only patient still in hospital has recovered from COVID-19. Importantly, our study provides detailed information on complications and presumed modes of death. This detailed analysis reveals that in the course of prolonged respiratory support a range of serious and outcome-relevant complications arise. The observed pattern with multi-organ failure implicates that COVID-19, at least in critically ill patients, should be regarded as a multi-system disease that reaches far beyond the respiratory tract and severe ARDS. This is in line with recent reports on endothelial cell involvement and diffuse vascular organ changes [16,17]. Further investigations including histopathological analysis of organ biopsies (ante- and post-mortem) are needed to elucidate critical organ involvement, as well as underlying pathophysiological mechanisms. The high rate of thromboembolic complications corroborates recent findings in case series and autopsy studies of a pronounced coagulopathy in severe COVID-19 [18–21]. The observed high incidence of septic shock possibly contributed to a compromised microcirculation, but may also be a consequence thereof. However, given the severity of COVID-19 in the ICU subgroup (indicated by the high proportion of moderate and severe ARDS, low Horovitz indices and the high rate of complications) it is noteworthy that 1 out of 2 ICU patients was discharged alive.

In the context of COVID-19, randomised controlled trials cannot be realised for all treatment modalities (pharmacological or supportive). Therefore data of observational studies will need to be analysed and compared [22,23]. In the current study we take advantage of a multistate model analysis [13]. This approach provides insights into time-dynamic effects and clinical outcomes, avoids common survival biases, and acknowledges active cases by taking into account censoring. In addition to the predicted probabilities for discharge and death, expected average durations in hospital can be calculated for the different states [24]. Visualisation using a stacked probability plot provides easy-to-interpret, yet compact and comprehensive information on the patients' clinical progress. This is in line with the proposals of the WHO and the COMET initiative regarding endpoints in clinical COVID-19 studies [25]. By applying such a multistate analysis our study provides firm estimates of in-hospital mortality rates and allows a more precise calculation of required ICU and ECMO capacities and therefore allocation of resources in a given care model [26].

Our study has limitations, primarily those inherent to its retrospective observational design. It is a monocenter study, which may limit generalisability. Yet the monocentric design may be considered a prerequisite to study treatment results in a specific care model at an experienced ARDS center. The limited number of patients precluded an analysis of specific treatment strategies, both in terms of antiviral or anti-inflammatory agents, anticoagulation strategies, and time-sensitive supportive strategies. While the primary endpoint of in-hospital death is reliably determined retrospectively, uncertainties remain in evaluating the mode of death. We tried to minimize this uncertainty by performing individual case review by two independent experienced physicians and explicitly avoiding causal assumptions.

## Conclusions

In summary, our study delineates that even under non-restricted care conditions COVID-19-related morbidity and mortality is high, especially in patients needing ICU management. Beside the search for potent antiviral agents, future research efforts should focus on strategies to alleviate or prevent complications identified in our study. Moreover, our findings underline the need for continued efforts in preventive measures and development of an effective vaccine. Finally, we demonstrate that by using a multistate model solid estimates for required ICU and ECMO capacities can be provided. Therefore, this work exemplifies, how best to report on COVID-19 studies to allow for meaningful comparisons of different treatment and care modalities.

## Supporting information

**S1 Fig. COVID dynamic care model of the University Medical Center Freiburg.** Patient flow in the dynamic care model established by the Task force Coronavirus (consisting of representatives of the ID department, Emergency department, Virology and Infection control Departments and the Pandemic Operational Committee of the University Medical Center Freiburg): Patients from the outpatient setting or inter-hospital tranfers were evaluated in dedicated areas in the emergency department. Confirmed COVID patients were distributed according to severity of disease on regular wards with or without monitoring. Patients with suspicion of COVID were admitted to separate holding areas. Unstable patients, ICU transfers or admissions to the ECMO facility were managed via the ICU coordinator and allocated to dedicated ICU and ECMO facilities. The dynamic care model included an escalation strategy, in which additional regular wards and ICU beds were equipped, physicians and nursing staff were trained and these wards were subsequently recruited upon utilisation of a certain threshold of COVID bed capacities. ID Infectious diseases, ICU Intensive care unit, IMC Intermediate care ward, COVID Coronavirus Disease 2019.
(PNG)

**S2 Fig. COVID response at the University Medical Center Freiburg.** The measures implemented in the COVID-19 response, the evolution of the peak incidences in the region (COVID-19 cases/100.00/day [dates of registration at local health authorities]) and the corresponding number of admissions in the University Medical Center Freiburg.
(PNG)

**S3 Fig. Schematic diagram of the applied multistate model.** The six state model considers the events hospitalisation in 1) regular ward, 2) ICU, 3) mechanical ventilation (MV), 4) ECMO, 5) discharge and 6) death. The boxes represent the possible states a patient may encounter and the arrows represent the possible transitions from one state to another. Thus, the arrows between the states show which transitions are possible.
(PNG)

**S4 Fig. Stacked probability plots for the multistate model stratified by age.** Stacked probability plots for the multistate model stratified by age. The plots illustrate in more detail the results of the competing risks regression models (however, not adjusted for other covariates). The graphs indicates that older patients have an increased risk to stay longer in hospital, to be admitted to the ICU, to need mechanical ventilation (including for a longer duration), and to die in hospital.
(PNG)

**S5 Fig. Stacked probability plots for the multistate model stratified by sex.** Stacked probability plots for the multistate model stratified by sex. The plots illustrate in more detail the results of the competing risks regression models (however, not adjusted for other covariates). The graphs indicates that male patients have an increased risk to be admitted to the ICU, to need mechanical ventilation, and to die in hospital.
(PNG)

**S6 Fig. Stacked probability plots for the multistate model stratified by the presence of comorbidities.** Stacked probability plots for the multistate model stratified by the presence of comorbidities. The plots illustrate in more detail the results of the competing risks regression models (however, not adjusted for other covariates). The graphs indicates that patients with one or more comorbidities have an increased risk to stay longer in hospital, to be admitted to the ICU, to need mechanical ventilation, and to die in hospital.
(PNG)

**S7 Fig. Stacked probability plots for the multistate model stratified by the presence of immunodeficiency.** Stacked probability plots for the multistate model stratified by the presence of immunodeficiency. The plots illustrate in more detail the results of the competing risks regression models (however, not adjusted for other covariates). The graphs indicates that immunodeficient patients have an increased risk to stay longer in hospital, yet, a decreased risk to be admitted to the ICU, to need mechanical ventilation, and to die in hospital.
(PNG)

**S8 Fig. Stacked probability plots for the multistate model stratified by presence of malignancy/neoplasm.** Stacked probability plots for the multistate model stratified by presence of malignancy/neoplasm. The plots illustrate in more detail the results of the competing risks regression models (however, not adjusted for other covariates). The graphs indicates that patients with malignancies or neoplasms have a slightly increased risk to be admitted to the ICU and to need mechanical ventilation, yet, no increased risk to die in hospital.
(PNG)

**S1 Data. R-code for data analysis.**
(HTML)

## Acknowledgments

### Declarations

(With linked authorship to members of the COVID UKF Study Group).

We thank all members of the COVID UKF Study Group who contributed to the development and implementation of the dynamic care model and were involved in patient care, virological diagnostics or infection control: Gabriele Peyerl-Hoffmann, Stephan Horn, Daniel Hornuss, Katharina Laubner, Dominik Bettinger, Christoph Jäger, Eric Peter Prager, Viviane Zotzmann, Dawid L. Staudacher, Cornelius Waller, Hans Fuchs, Sebastian Fähndrich, Hans-Jörg Busch, Monika Engelhardt, Hartmut Bürkle, Michael Berchtold-Herz, Thorsten Hammer, Felix Hans, Marcus Panning, Hartmut Hengel, Peter Hasselblatt, Wolfgang Kühn, Daniel Duerschmied, Robert Thimme, Christoph Bode, Hajo Grundmann, Philipp Henneke.

## Author Contributions

**Conceptualization:** Siegbert Rieg, Martin Wolkewitz, Winfried V. Kern, Paul Biever.

**Data curation:** Maja von Cube, Johannes Kalbhenn, Stefan Utzolino, Katharina Pernice, Lena Bechet, Johanna Baur, Corinna N. Lang, Dirk Wagner, Martin Wolkewitz, Paul Biever.

**Formal analysis:** Siegbert Rieg, Maja von Cube, Martin Wolkewitz, Winfried V. Kern.

**Investigation:** Siegbert Rieg, Johannes Kalbhenn, Stefan Utzolino, Katharina Pernice, Lena Bechet, Johanna Baur, Corinna N. Lang, Dirk Wagner, Winfried V. Kern, Paul Biever.

**Methodology:** Siegbert Rieg, Maja von Cube, Martin Wolkewitz.

**Project administration:** Lena Bechet.

**Supervision:** Siegbert Rieg, Paul Biever.

**Validation:** Siegbert Rieg, Maja von Cube, Lena Bechet, Johanna Baur, Martin Wolkewitz, Winfried V. Kern, Paul Biever.

**Visualization:** Siegbert Rieg.

**Writing – original draft:** Siegbert Rieg, Maja von Cube.

**Writing – review & editing:** Siegbert Rieg, Maja von Cube, Johannes Kalbhenn, Stefan Utzolino, Katharina Pernice, Lena Bechet, Johanna Baur, Corinna N. Lang, Dirk Wagner, Martin Wolkewitz, Winfried V. Kern, Paul Biever.

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
