## [Decision Letter · Decision Letter 0]

18 Sep 2020

PONE-D-20-25207

COVID-19 in-hospital mortality and mode of death in a dynamic and non-restricted tertiary care model in Germany

PLOS ONE

Dear Dr. Rieg,

Thank you for submitting your manuscript to PLOS ONE. After careful consideration, we feel that it has merit but does not fully meet PLOS ONE’s publication criteria as it currently stands. Therefore, we invite you to submit a revised version of the manuscript that addresses the points raised during the review process.

We look forward to receiving your revised manuscript.

Kind regards,

Andrea Ballotta

Academic Editor

PLOS ONE

Journal Requirements:

Additional Editor Comments:

Thank you very much for your contribution. The paper is of great interest but it needs some minor issues to be answered.

Reviewers' comments:

Reviewer's Responses to Questions

**Comments to the Author**

1. Is the manuscript technically sound, and do the data support the conclusions?

Reviewer #1: Yes

Reviewer #2: Yes

2. Has the statistical analysis been performed appropriately and rigorously? 

Reviewer #1: Yes

Reviewer #2: Yes

3. Have the authors made all data underlying the findings in their manuscript fully available?

Reviewer #1: Yes

Reviewer #2: No

4. Is the manuscript presented in an intelligible fashion and written in standard English?

Reviewer #1: Yes

Reviewer #2: Yes

5. Review Comments to the Author

Reviewer #1: Review of the article entitled “COVID-19 in hospital mortality and mode of death…”

The Article addresses a clue topic nowadays. It has the pride to be well designed and to be methodologically appropriate for the aims. The format is clinical and observational retrospective. Although done in a single hospital, the study population is large with the advantage to use an already well entrained network of expertise. This special point is of value when the impact of comorbidities on the survival rate is addressed as well as when the cause of fatal events is assessed.

Analytical revision

Abstract

Line 3: “optimized care conditions”. I suggest “state of the art care”. Needless to say, the search for stratification of the clinical severity and of the treatment are ongoing.

Line 23: ”substantial”. I suggest “very high”.

Text

88-97: The concept of “dynamic care model” in a well-equipped Hospital, that permits to assign patients without restriction, should be more briefly summarized.

115. ”(outlined in Supplementary Figure S 1)”. The Figure S 1 is unessential to address the three aims of the study

Line 118-119: “or being involved via the ID consultation service”. Unessential

119-121: The Supplementary Figure S 2 is interesting in speculative terms since adds information on the real word. It is unessential to address the aims of the study.

140. “Supplementary Table1”. The information given by the Suppl. Table 1 is essential to the value of the study. My suggestion is to include it straightforward in the results section rather than in the method section.

148. “designated each death to, related to COVID-!9 or unrelated to…”. Sentence to be reshaped .

162-163. “The model is shown in Supplementary Figure 3 “. Figure Suppl.3 is unessential.

167-169. both sentences are unessential in the “statistical analysis “paragraph.

209-210. “Supplementary Figure S 4”, although the time span of in hospital stay is of importance, the Figure is complex and similar information may be deducted from the remaining Figures.

229-230. “Supplementary Figures S5-S9”. These figures give great value to the overall work. At a glance the reader can appreciate 1) the prognosis over time according to the class of entry and 2) the chance to switch into a different state. As such I recommend to insert those Figures in the final text. Only the Suppl. Figure S6 does not seem essential, so it can be removed in order to reduce the load. I recommend to rephrase completely the legends by putting in clear the clinical and prognostic meaning beyond the statistical technicality. For instance, the Suppl. Figure 7 could be usefully inserted at line 222-223.

255. ”time points of death are depicted in the cohort plot in Supplementary Figure S4”: as already told , the Figure S4 is overcrowded . It can be removed.

267. “, and on the required health service resources,”: the specification breaks the main message. It could be deleted.

270-272. All the sentence may be rephrased by saying how the information on mortality in ICU patients is strongly required by medical and social institutions.

286: “usage of”: look to a better form.

288-289. The information about the relevance of comorbidities on mortality is essential to the topic. The results of the study are clearly at a difference with the current narration. This point deserves adequate.

The Figure S 8 should be quoted.

303-306. “however…..alive”: the message is clear, although the sentence needs revision.

319-326. “monocenter study”: I do not see a limitation, since homogeneity of decision making is of value in such a complex context. “generalisability” is not mandatory today. ”antiviral or anti-inflammatory agents”: 1. Also anticoagulation is dependent upon clinical sense or imposed by cannulation of the vessels. Not to say the challenge between intravascular thrombosis and life-threatening bleeding.

2. as such a brief comment on the uncertainity of starting anticoagulation could be of help to the reader. In this paragraph or previously.

327. Conclusions: the paragraph sound more as sum up than true assertive conclusions. Since the matter is hot and the study is rich of information, a new draft is recommended.

437.Table 1. “NEWS2 score”: the acronym should be put in extenso in line 439.

“CRP”: same comment. The range of normality has to be added to the first column “PCT”: same comment. The range of normality has to be added to the first column “IL-6”: same comment. The range of normality has to be added to the first column.

The range of normality for D-dimers has also to be added.

Since the occurrence of intravascular thrombosis and of life-threatening bleeding were so high in the study population, at least the count of the platelets should be included in order to infer potential consumption.

Medical treatment: Heparin and Corticosteroids should be included in the list, if employed.

460. Table 4: “Cardiac arrest – CPR w/o ROSC”: the acronyms should be put in extenso at the bottom. Infact cardiac arrest is clinically clear and further specifications could be unessential.

478-479: Figure 2. The numerosity of each group should be written as (n=….). It is very needed since the MV group is obviously included in the ICU group.

Reviewer #2: COVID-19 in-hospital mortality and mode of death in a dynamic and non-restricted tertiary care model in Germany

The authors provide a retrospective study of COVID-19 patients care at the University Medical Center Freiburg during the peak of the epidemic in March/April 2020. This is an important study for the research community as much can be learned from the experience of medical doctors and scientists that were in direct contact with patients, in terms of survival rates, treatment and hospital management. Additional factors that make this study interesting are that the Medical Center had at its disposal the latest available treatments and knowledge, and had enough space in its ICU unit so that it was not overwhelmed by the number of patients in critical conditions.

The authors use a multistate analysis, where a patient moves through different states before being released healthy or passing away, which allows them to understand how different categories of patients (elderly vs relatively younger, male vs female) are likely to progress once they enter the Medical Center. They also use a survival analysis to demonstrate that elderly and male patients are more likely to have complications.

Overall, the paper is very well written, however we think that improvements are necessary before we can recommend it for publication.

Most importantly, more explanation of the methodology used is necessary to ensure the clarity and reproducibility of the results, especially because this article is likely to reach a wide audience that is not restricted to experts in survival analysis. Also it seems that only summary data are provided in tables, which it seems not to be enough to comply with the full data availability statement.

Major points:

• In methods, it is mentioned that a multistate analysis is used, but no reference is given, nor details of what mathematical model was used and how it was implemented. Was the multistate model a Markov Chain? What are the assumptions here? How were the state change rates estimated?

• It is also mentioned that a risk factor analysis (competing risk model) is used, where the different states of hospitalisation were not differentiated, but also in this case no much explanation is given. Are modelling assumptions met? (e.g. proportionality of hazards). If a multistate survival analysis was used already (where basically each transition rate is a survival model), why use a different survival model here, where states are not considered?

• Fine and Gray model was used in the results but not discussed in the methods. Please provide more information about this method, why it was employed and what assumptions it required.

• In general, I would like to see more details of methods, such as what mathematical models were used, what assumptions, if the assumptions were met, citing appropriate literature, explaining why these method and not other competing methods were used.

• Another major point missing is whether there was anything that could be learned from treatments? How did they affect the path to discharge or death of the patients? Were they taken into consideration as confounding variables of the models? It would be useful to add at least a remark that this was investigated, even if the results of the effect of treatment were unconclusive.

• Code should be available for reproducibility for example by upload to a github account. At line 177, Rstudio version is not informative, it is just an advanced text editor for R code, authors should report which version of R was used and which R packages.

• I believe all data points should be available, not only summary data. The authors stated that all data were available but I was only able to find summary data. Could you provide a text file on github or an excel file as supplementary data?

• Is it possible to provide or estimate the false positive rate of the PCR-based test used to determine the SARS-CoV-2 infection? It could be interesting/relevant to know.

Minor points:

• Line 49, please spell out ARDS and ECMO in the abstract and put the abbreviation in parenthesis. Please do this for all abbreviation when used the first time.

• Line 86 I would remove “e.g.” as it seems unnecessary

• Line 305 “1 out of two ICU patients could was discharged alive”, remove “could” or change “could was” into “could be”.

• Line 307, please add a comma after “COVID-19”

6. PLOS authors have the option to publish the peer review history of their article (what does this mean?). If published, this will include your full peer review and any attached files.

Reviewer #1: No

Reviewer #2: No

---

## [Author Response · Author response to Decision Letter 0]

26 Oct 2020

Journal Requirements:

Reply: We tried to fulfill all format requirements.

Reply: Was corrected.

Additional Editor Comments:

Thank you very much for your contribution. The paper is of great interest but it needs some minor issues to be answered.

Reviewers' comments:

Reviewer's Responses to Questions

Comments to the Author

1. Is the manuscript technically sound, and do the data support the conclusions?

Reviewer #1: Yes

Reviewer #2: Yes

2. Has the statistical analysis been performed appropriately and rigorously? 

Reviewer #1: Yes

Reviewer #2: Yes

3. Have the authors made all data underlying the findings in their manuscript fully available?

Reviewer #1: Yes

Reviewer #2: No 

Due to the German Federal Data Protection Act (Bundesdatenschutzgesetz) and the fact that the Institutional Review Board of the University Medical Center Freiburg granted publication of only anonymised data, inclusion of the complete dataset is not possible. We included a statement in the Ethical Consideration section (see also in reply to Reviewer #2). 

4. Is the manuscript presented in an intelligible fashion and written in standard English?

Reviewer #1: Yes

Reviewer #2: Yes

5. Review Comments to the Author

Reviewer #1: Review of the article entitled “COVID-19 in hospital mortality and mode of death…”

The Article addresses a clue topic nowadays. It has the pride to be well designed and to be methodologically appropriate for the aims. The format is clinical and observational retrospective. Although done in a single hospital, the study population is large with the advantage to use an already well entrained network of expertise. This special point is of value when the impact of comorbidities on the survival rate is addressed as well as when the cause of fatal events is assessed.

Analytical revision

Abstract

Line 3: “optimized care conditions”. I suggest “state of the art care”. Needless to say, the search for stratification of the clinical severity and of the treatment are ongoing.

Line 23: ”substantial”. I suggest “very high”.

Reply: Both corrected.

Text

Lines 88-97: The concept of “dynamic care model” in a well-equipped Hospital, that permits to assign patients without restriction, should be more briefly summarized.

Reply: We shortened the respective paragraph. 

Line 115. ”(outlined in Supplementary Figure S 1)”. The Figure S 1 is unessential to address the three aims of the study.

Reply: We agree that the graph is not essential with regard to answering the primary questions addressed – that’s why we decided to move the figure into the supplementary material. However, as the organisational matters are of interest to other institutions (we already received several comments/requests) we would prefer to keep the Figures S1 and S2 included in the supplementary material. 

Line 118-119: “or being involved via the ID consultation service”. Unessential

Reply: Deleted.

119-121: The Supplementary Figure S 2 is interesting in speculative terms since adds information on the real word. It is unessential to address the aims of the study.

Reply: see comment above (line 115).

140. “Supplementary Table1”. The information given by the Suppl. Table 1 is essential to the value of the study. My suggestion is to include it straightforward in the results section rather than in the method section.

Reply: As suggested we included the former Suppl. Table 1 now as Table 5 in the main manuscript. 

148. “designated each death to, related to COVID-19 or unrelated to…”. Sentence to be reshaped .

Reply: Done.

162-163. “The model is shown in Supplementary Figure 3 “. Figure Suppl.3 is unessential.

Reply: For those that are not abreast of multistate analyses, the figure is of great help in illustrating the possible transitions between the different states. We therefore would like to keep Supplementary Figure 3 included in the supplementary material. 

167-169. both sentences are unessential in the “statistical analysis “paragraph.

Reply: Both sentences were deleted.

209-210. “Supplementary Figure S 4”, although the time span of in hospital stay is of importance, the Figure is complex and similar information may be deducted from the remaining Figures.

Reply: Supplementary Figure S 4 was deleted.

229-230. “Supplementary Figures S5-S9”. These figures give great value to the overall work. At a glance the reader can appreciate 1) the prognosis over time according to the class of entry and 2) the chance to switch into a different state. As such I recommend to insert those Figures in the final text. Only the Suppl. Figure S6 does not seem essential, so it can be removed in order to reduce the load. I recommend to rephrase completely the legends by putting in clear the clinical and prognostic meaning beyond the statistical technicality. For instance, the Suppl. Figure 7 could be usefully inserted at line 222-223.

Reply: Beside the strongest predcitor age, sex is a consistently found risk factor for a worse outcome - we do not see why this graph is less informative than the others. We agree that the stacked probability plots are very instructive, however, we are concerned that there will be too many figures in the main manuscript, if we move Figures S5-S9 in the final text. We would of course be happy to follow Editorial guidance on this.

As suggested we modified the figure legends and now delineate the clinical and prognostic meaning of the stacked probability plots.

255. ”time points of death are depicted in the cohort plot in Supplementary Figure S4”: as already told , the Figure S4 is overcrowded . It can be removed.

Reply: Supplementary Figure S 4 was deleted.

267. “, and on the required health service resources,”: the specification breaks the main message. It could be deleted.

Reply: Done. 

270-272. All the sentence may be rephrased by saying how the information on mortality in ICU patients is strongly required by medical and social institutions.

Reply: We restructured the sentence and included the information. 

286: “usage of”: look to a better form.

Reply: Was replaced by ‚application of‘. 

288-289. The information about the relevance of comorbidities on mortality is essential to the topic. The results of the study are clearly at a difference with the current narration. This point deserves adequate. The Figure S 8 should be quoted.

Reply: We included a specific remark on the impact of immunodefciency in the discussion (lines 340-341 in track change modus version). 

303-306. “however…..alive”: the message is clear, although the sentence needs revision.

Reply: The mistake was corrected. 

319-326. “monocenter study”: I do not see a limitation, since homogeneity of decision making is of value in such a complex context. “generalisability” is not mandatory today. ”antiviral or anti-inflammatory agents”: 1. Also anticoagulation is dependent upon clinical sense or imposed by cannulation of the vessels. Not to say the challenge between intravascular thrombosis and life-threatening bleeding. 2. as such a brief comment on the uncertainity of starting anticoagulation could be of help to the reader. In this paragraph or previously.

Reply: We agree with this notion and already point out, that the monocenter design is needed if specific levels or models of care are studied. Moreover, we included anticoagulation as a potential but so far undefined management strategy (line 375 in track change modus version). 

327. Conclusions: the paragraph sound more as sum up than true assertive conclusions. Since the matter is hot and the study is rich of information, a new draft is recommended.

Reply: We thank the reviewer for this comment and modified the conclusions accordingly. We emphasise now on the consequences and future research needs that arise from the results of our study. 

437. Table 1. “NEWS2 score”: the acronym should be put in extenso in line 439.

“CRP”: same comment. The range of normality has to be added to the first column “PCT”: same comment. The range of normality has to be added to the first column “IL-6”: same comment. The range of normality has to be added to the first column. The range of normality for D-dimers has also to be added.

Reply: The requested information was added.

Since the occurrence of intravascular thrombosis and of life-threatening bleeding were so high in the study population, at least the count of the platelets should be included in order to infer potential consumption.

Reply: The requested information was added.

Medical treatment: Heparin and Corticosteroids should be included in the list, if employed.

Reply: We here report patients from the first 2,5 months of the pandemic. Data from trials investigating the impact of corticosteroids (e.g. RECOVERY trial) were not available yet, thus, no patient received corticosteroids with COVID-19/COVID-19-associated ARDS as indication. Concerning heparin, all hospitalised patients received low-molecular weight (or heparin if the former was contraindicated) in the usual prophylaxis dose. Those patients with confirmed thrombembolic events received therapeutic anticoagulation, however, there was no systematic use of an anticoagulation strategy. 

460. Table 4: “Cardiac arrest – CPR w/o ROSC”: the acronyms should be put in extenso at the bottom. Infact cardiac arrest is clinically clear and further specifications could be unessential.

Reply: The requested information was added.

478-479: Figure 2. The numerosity of each group should be written as (n=….). It is very needed since the MV group is obviously included in the ICU group.

Reply: The requested information was added in Figure 2.

Reviewer #2: COVID-19 in-hospital mortality and mode of death in a dynamic and non-restricted tertiary care model in Germany

The authors provide a retrospective study of COVID-19 patients care at the University Medical Center Freiburg during the peak of the epidemic in March/April 2020. This is an important study for the research community as much can be learned from the experience of medical doctors and scientists that were in direct contact with patients, in terms of survival rates, treatment and hospital management. Additional factors that make this study interesting are that the Medical Center had at its disposal the latest available treatments and knowledge, and had enough space in its ICU unit so that it was not overwhelmed by the number of patients in critical conditions.

The authors use a multistate analysis, where a patient moves through different states before being released healthy or passing away, which allows them to understand how different categories of patients (elderly vs relatively younger, male vs female) are likely to progress once they enter the Medical Center. They also use a survival analysis to demonstrate that elderly and male patients are more likely to have complications.

Overall, the paper is very well written, however we think that improvements are necessary before we can recommend it for publication.

Most importantly, more explanation of the methodology used is necessary to ensure the clarity and reproducibility of the results, especially because this article is likely to reach a wide audience that is not restricted to experts in survival analysis. Also it seems that only summary data are provided in tables, which it seems not to be enough to comply with the full data availability statement.

Major points:

• In methods, it is mentioned that a multistate analysis is used, but no reference is given, nor details of what mathematical model was used and how it was implemented. Was the multistate model a Markov Chain? What are the assumptions here? How were the state change rates estimated?

Reply: We included the information that a Markovian multistate analysis was used. A reference is given with regard to the applied statistcal methodolgy and the required assumptions (Lines 160-165 in track change modus version).

• It is also mentioned that a risk factor analysis (competing risk model) is used, where the different states of hospitalisation were not differentiated, but also in this case no much explanation is given. Are modelling assumptions met? (e.g. proportionality of hazards). If a multistate survival analysis was used already (where basically each transition rate is a survival model), why use a different survival model here, where states are not considered?

Reply: The simplified model was used to avoid biases from confounder treatment feedback. We wrote in the main manuscript: “For the risk factor analysis, we used a competing risks model to study effects on the time from hospitalisation to death in the hospital. To avoid collider bias, in this model, the different states of hospitalisation (regular ward, ICU, MV, ECMO) were not differentiated” (Lines 173-175 in track change modus version).

• Fine and Gray model was used in the results but not discussed in the methods. Please provide more information about this method, why it was employed and what assumptions it required.

Reply: We precisized in the main manuscript (Lines 177-180 in track change modus version): “Then, we estimated the subdistribution hazard ratio of death using a Fine and Gray model. The subdistribution hazard ratio quantifies the effect of risk factors on the absolute risks (rather than the rates) thereby combining the direct and indirect effects found in the cause-specific analysis.”

• In general, I would like to see more details of methods, such as what mathematical models were used, what assumptions, if the assumptions were met, citing appropriate literature, explaining why these method and not other competing methods were used.

Reply: As there is already a lot of methodological and technical information in the manuscript, we decided to give those interested in the methodology an up-to-date reference that includes a detailed summary of the chosen approach [Hazard D et al, Ref. 22], please see also lines 162-165: “Multistate model analysis has not only the major advantage that the time dyamics of a patients disease progression are taken into account but also that multiple events are studied simulanteuously. The model is shown in Fig S3. The statistical methodlogy and required assumptions are outlined in detail in (14).“

• Another major point missing is whether there was anything that could be learned from treatments? How did they affect the path to discharge or death of the patients? Were they taken into consideration as confounding variables of the models? It would be useful to add at least a remark that this was investigated, even if the results of the effect of treatment were unconclusive.

Reply: As mentionned above we report on patients from the first 2,5 months of the pandemic. No specific antiviral agent (such as remdesivir) was available at that time. Accordingly, results of the RECOVERY trial concerning potential benefits of patients treated with dexamthasone were not published yet, thus, no patient received corticosteroids with COVID-19/COVID-19-associated ARDS as indication. Moreover, as several randomized trials did not find any effects of lopinavir/ritonavir or hydroxychloroquin/chloroquine (and as the number of patients treated with these agents was too small), we refrained from performing in depth analyses with regard to pharmacological treatment strategies. This limitation is outlined in the discussion (lines 374-376). 

• Code should be available for reproducibility for example by upload to a github account. At line 177, Rstudio version is not informative, it is just an advanced text editor for R code, authors should report which version of R was used and which R packages.

Reply: All analyses were performed with R Version 4.0.2. This information is provided in statistical methods. Moreover, the R-code for data analysis is now included in the supplementary material.

• I believe all data points should be available, not only summary data. The authors stated that all data were available but I was only able to find summary data. Could you provide a text file on github or an excel file as supplementary data?

Reply: Due to the German Federal Data Protection Act (Bundesdatenschutzgesetz) and the fact that the Institutional Review Board of the University Medical Center Freiburg granted publication of only anonymyzed data, inclusion of the complete dataset is not possible. We included a statement in the Ethical Consideration section. 

• Is it possible to provide or estimate the false positive rate of the PCR-based test used to determine the SARS-CoV-2 infection? It could be interesting/relevant to know.

Reply: The specificity of the PCR-test used (RealStar SARS-CoV-2 RT-PCR kit 1.0 (Altona Diagnostics, Hamburg, Germany) is considered to be >99%. 

Minor points:

• Line 49, please spell out ARDS and ECMO in the abstract and put the abbreviation in parenthesis. Please do this for all abbreviation when used the first time.

Reply: The abbreviations were introduced now throughout the manuscript. 

• Line 86 I would remove “e.g.” as it seems unnecessary

Reply: Was removed.

• Line 305 “1 out of two ICU patients could was discharged alive”, remove “could” or change “could was” into “could be”.

• Line 307, please add a comma after “COVID-19”.

Reply: Both mistakes were corrected. 

6. PLOS authors have the option to publish the peer review history of their article (what does this mean?). If published, this will include your full peer review and any attached files.

Do you want your identity to be public for this peer review? For information about this choice, including consent withdrawal, please see our Privacy Policy.

Reviewer #1: No

Reviewer #2: No

---

## [Editor Report · Decision Letter 1]

28 Oct 2020

COVID-19 in-hospital mortality and mode of death in a dynamic and non-restricted tertiary care model in Germany

PONE-D-20-25207R1

Dear Dr. Rieg,

We’re pleased to inform you that your manuscript has been judged scientifically suitable for publication and will be formally accepted for publication once it meets all outstanding technical requirements.

Kind regards,

Andrea Ballotta

Academic Editor

PLOS ONE

Additional Editor Comments (optional):

Congratulations your manuscript "COVID-19 in-hospital mortality and mode of death in a dynamic and non-restricted tertiary care model in Germany" is suitable for publication.
---

## [Editor Report · Acceptance letter]

3 Nov 2020

PONE-D-20-25207R1 

COVID-19 in-hospital mortality and mode of death in a dynamic and non-restricted tertiary care model in Germany 

Dear Dr. Rieg:

I'm pleased to inform you that your manuscript has been deemed suitable for publication in PLOS ONE. Congratulations! Your manuscript is now with our production department. 

Kind regards, 

on behalf of

Dr. Andrea Ballotta 

Academic Editor

PLOS ONE